# Flexible interlocked porous frameworks allow quantitative photoisomerization in a crystalline solid

Yongtai Zheng [1], Hiroshi Sato [2], Pengyan Wu[1], Hyung Joon Jeon[1], Ryotaro Matsuda[1,3,5] & Susumu Kitagawa[1,4]

Photochromic molecules have shown much promise as molecular components of stimuli-responsive materials, but despite recent achievements in various photoresponsive materials, quantitative conversion in photochemical reactions in solids is hampered by the lack of intrinsic structural flexibility available to release stress and strain upon photochemical events. This issue remains one of the challenges in developing solid-state photoresponsive materials. Here, we report a strategy to realize photoresponsive crystalline materials showing quantitative reversible photochemical reactions upon ultraviolet and visible light irradiation by introducing structural flexibility into crystalline porous frameworks with a twofold interpenetration composed of a diarylethene-based ligand. The structural flexibility of the porous framework enables highly efficient photochemical electrocyclization in a single-crystal-to-single-crystal manner. $CO_2$ sorption on the porous crystal at 195 K is reversibly modulated by light irradiation, and coincident X-ray powder diffraction/sorption measurements clearly demonstrate the flexible nature of the twofold interpenetrated frameworks.

[1] Institute for Integrated Cell-Material Sciences (WPI-iCeMS), Kyoto University, Katsura, Nishikyo-ku, Kyoto 615-8510, Japan. [2] Department of Chemistry and Biotechnology, School of Engineering, The University of Tokyo, 7-3-1 Hongo, Bunkyo-ku, Tokyo 113-8656, Japan. [3] Japan Science and Technology Agency (JST), PRESTO, 4-1-8 Honcho, Kawaguchi, Saitama 332-0012, Japan. [4] Department of Synthetic Chemistry and Biological Chemistry, Graduate School of Engineering, Kyoto University, Katsura, Nishikyo-ku, Kyoto 615-8510, Japan. [5] Present address: Department of Applied Chemistry, Graduate School of Engineering Nagoya University Furo-cho, Chikusa-ku, Nagoya 464-8603, Japan. Correspondence and requests for materials should be addressed to H.S. (email: hsato@macro.t.u-tokyo.ac.jp) or to S.K. (email: kitagawa@icems.kyoto-u.ac.jp)

Photochromic molecules are substances showing colour changes upon photoirradiation, which are accompanied by changes of geometric structures, electronic states, and chemical and physical properties[1]. Such chemicals have played key roles in developing various photoresponsive materials, including polymers, gels, and liquid crystals, and the demand for such materials is growing in various fields, including medicinal chemistry[2–5]. Additionally, porous coordination polymers (PCPs) or metal–organic frameworks (MOFs), have attracted much attention as a new class of porous materials with crystalline frameworks[6–9]. Taking advantage of the modular synthesis of the frameworks from metal ions and organic ligands, various types of functionalities (e.g., acid/base sites, hydrogen bonding sites, and chiral sites) have been successfully introduced into the frameworks with unique porous properties. Furthermore, photochemically reactive modules have also been introduced on the pore surfaces as struts or in the pores of PCPs/MOFs as guest molecules to control the porous properties by light[10–16], potentially leading to on-demand type materials for molecular storage, separation, and catalysis.

The keys to achieving reversible control of the porous functions of PCPs/MOFs by light with photochromic ligands are the following. First, the ligand shows high efficiency in the photoisomerization reactions between the isomers. Second, the isomers are sufficiently thermally stable to not be converted into another isomer in the dark. Otherwise, undesired thermal backward reactions or side reactions would be operative. Third, the photochromic unit must have high fatigue resistance for high repeatability. Promising candidates as photoresponsive modules that fullfil all the requirements are dithienylethene (DTE) derivatives[17]. DTE derivatives show reversible photocyclization between open- and closed-ring isomers upon ultraviolet (UV) and visible light irradiation. Because of these prominent characteristics, DTE derivatives have been extensively studied in the field of responsive materials[18]. Recently, some pioneering works on photoresponsive MOFs composed of DTE-type ligands have been reported[19–24]. However, DTE ligands in crystalline states exhibited low photoisomerization efficiencies (or photochemical conversion was not determined), and the phenomena were far from ideal, despite the high conversion of the ligand itself in solution. This might be due to the effect of the packing in the rigid frameworks, where the change in orientation and volume required for the isomerization would not be tolerated[25]. It is necessary to introduce a different technique for the high conversion of light energy into photoisomerization in the solid or crystal state.

Hence, we designed a framework having flexibility[26–30], such that during photoisomerization, sufficient room is maintained for structural changes of DTE moieties for quantitative conversion, with the release of strain through the framework structural change upon photoisomerization. Accordingly, we have selected PCPs/MOFs with entangled frameworks showing dynamic behavior from relative positional changes of the constituent frameworks[31–35]. This flexible nature allowed us to develop a PCP showing highly effective isomerization and cooperative structural transformations by combining flexible frameworks with DTE derivatives. Herein, we report the synthesis and characterization of a photoresponsive PCP with a twofold interpenetrated framework composed of a DTE-based ligand. Taking advantage of the flexible nature of the entangled framework, the porous crystal shows a quantitative and reversible isomerization upon UV and visible light irradiation, which is applicable to reversible photomodulation of its gas sorption properties.

## Results

**Synthesis and characterization of the soft porous crystals.** A flexible porous crystal was prepared using a photochromic module, a bis(pyridyl)dithienylethene ($L_O$)[36, 37] that shows a highly effective interconversion between open- ($L_O$) and closed-ring ($L_C$) isomers (Fig. 1) in various solvents (Supplementary Fig. 1) and moderate reactivity in the solid state (~30%). PCP 1 was synthesized by a solvothermal reaction of $Zn(NO_3)_2 \cdot 6H_2O$, $L_O$, and 1,4-benzenedicarboxylic acid ($H_2bdc$) in $N,N$-dimethyl-formamide (DMF). The structure of 1 ($[Zn_4(bdc)_4(L_O)_2 \cdot 4DMF \cdot H_2O]_n$) was analyzed by single-crystal X-ray diffraction (Fig. 2a–c and Supplementary Fig. 2a–c). The paddlewheel-type zinc complexes (Supplementary Fig. 2a) composed of $bdc^{2-}$ and $Zn^{2+}$ are extended to 2D sheets (Supplementary Fig. 2b), which are further connected by the coordination of the pyridyl moieties in $L_O$ to afford a 3D pillared layer structure with an inter-layer distance of 21.6 Å between the 2D sheets (Fig. 2b and Supplementary Fig. 2c). The twofold interpenetrated framework affords a channel structure that accommodates DMF and water molecules; the void volume was evaluated by *PLATON* to be 30% of the total unit cell volume. The connected 2D channels running parallel to the

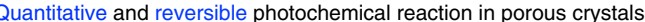

Quantitative and reversible photochemical reaction in porous crystals

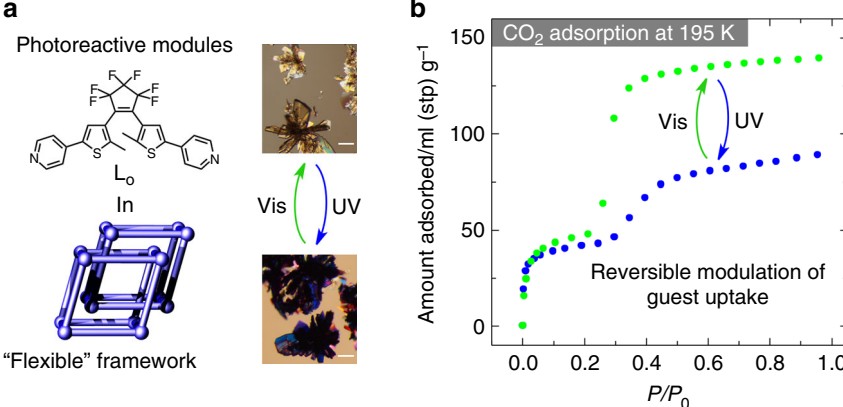

**Fig. 1** Quantitative and reversible photoisomerization in a flexible porous crystal for modulating $CO_2$ sorption. **a** The combination of a photoreactive module, a dithienyl-based ligand ($L_O$), and a twofold interpenetrated porous framework as a flexible platform achieves a quantitative and reversible photochemical reaction. **b** The highly effective photochemical reaction in the porous crystal realizes a reversible modulation of $CO_2$ sorption. The $CO_2$ adsorption isotherms were measured at 195 K. The microscope images show the colour change of the crystals upon light irradiation and the scale bars indicate 50 μm

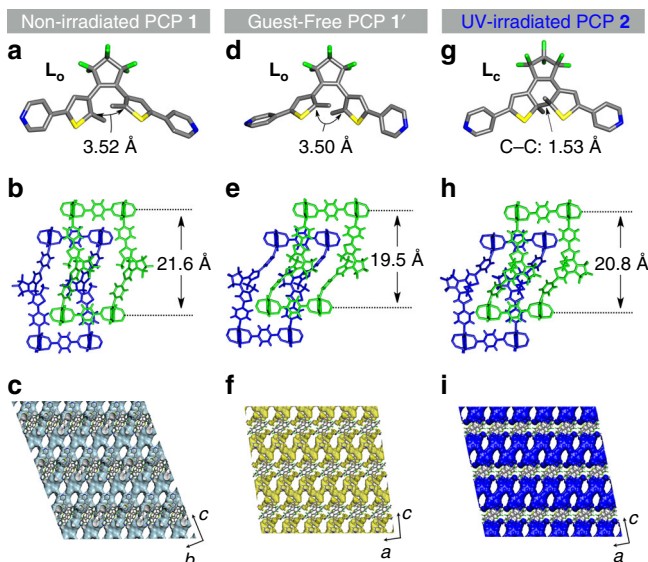

**Fig. 2** X-ray crystal structural analyses. Crystal structures of as-synthesized PCP **1** a–c, guest-free **1′** d–f, and UV-irradiated **2** g–i. Atoms are coloured as follows: Zn, *cyan*; C, *grey*; N, *blue*; O, *red*; S, *yellow*; F, *green*. Hydrogen atoms, DMF, and water molecules are omitted for clarity. **a**, **d**, **g** Photoreactive ligands (**L$_O$** or **L$_C$**) found in PCPs. **b**, **e**, **h** The twofold interpenetrated frameworks composed of a Zn paddle-wheel cluster, bdc$^{2-}$, and photoreactive ligands. **c** Three-dimensionally connected channels in **1**. The *pale blue curved surface* is the Connolly surface (Connolly radius: 1.6 Å) in **1**. **f** One-dimensional zig-zag channels in **1′**. The *yellow curved surface* is the Connolly surface (Connolly radius: 1.6 Å) in **1′**. **i** One-dimensional zig-zag channels in **2**. *The blue curved surface* is the Connolly surface (Connolly radius: 1.6 Å) in **2**

*bc*-plane (Fig. 2c) are crossed by another channel along the *a*-axis to form the 3D porous structure (Supplementary Fig. 3a). The photochromic units are arranged in a 1D fashion along the *a*-axis (Supplementary Fig. 4a), and the neighboring two photochromic ligands from different networks form a pair of the DTE moieties through π-π interactions at a distance of 3.7 Å (Supplementary Fig. 4b). Most importantly, the reactive carbon atoms in the thienyl groups were found to be located 3.52 Å apart in an anti-parallel conformation of the open-form **L$_O$** in **1** (Fig. 2a), which indicates feasibility for photocyclization reaction (<4.2 Å)[38].

Thermogravimetric analysis (TGA) showed that the guest molecules can be removed up to 140 °C to give the guest-free PCP **1′** ([Zn$_2$(bdc)$_2$(**L$_O$**)]$_n$) that is thermally stable up to ~330 °C (Supplementary Fig. 6). We determined the single-crystal X-ray structure of the guest-free PCP **1′** (Fig. 2d–f and Supplementary Fig. 2d–f) obtained by heating the crystals of **1** at 120 °C under reduced pressure. Although the network connectivity composed of Zn$^{2+}$ and ligands was intact, the pyridyl groups leaned against the equatorial planes of Zn$^{2+}$, resulting in the deformation of the framework and channel structures. The interlayer distance between the 2D sheets formed by Zn$^{2+}$ and bdc$^{2-}$ became shorter (19.5 Å) than that in the as-synthesized **1** (21.6 Å), which was accompanied by a lateral positional change of the sheets along the *a*-axis (Fig. 2b, e and Supplementary Fig. 2c, f). Compared with **1**, the channels are partially disconnected to afford a zig-zag 1D channel parallel to the *a*-axis in **1′** (Fig. 2f and Supplementary Fig. 3b). Thus, the void space was considerably reduced from 30% (as-synthesized PCP **1**) to 15% (guest-free PCP **1′**) of the total unit cell volumes. These structural transformations in the crystalline frameworks clearly showed the flexibility in the local and global structures of the PCP. It is worth noting that regardless of the variation of geometry of the photochromic units in **1′**, the

distance between the reactive carbon atoms was still 3.50 Å, which would allow the photochemical reaction to proceed (Fig. 2d).

**Single-crystal-to-single-crystal photochemical transformation of a soft porous crystal**. The flexible nature of the framework containing photochromic units encouraged us to investigate the photochemical reactivity of **L$_O$** and structural changes in its single crystals. The colourless crystals of **1** instantly turned dark blue upon exposure to UV light (Fig. 1), indicating the formation of the closed-ring isomer **L$_C$** in the crystals. By $^1$H nuclear magnetic resonance (NMR) spectroscopic analysis of the irradiated crystals digested in DMSO-$d_6$/aq. HCl at 298 K, we confirmed that the photochemical conversion from an **L$_O$** to **L$_C$** in the PCP was almost quantitative (>95%, Supplementary Fig. 8), which is comparable to the reaction of **L$_O$** in solution. The backward reaction from **L$_C$** to **L$_O$** by visible light irradiation was also confirmed as almost quantitative (Supplementary Fig. 9). It is noteworthy that there is only a limited report for such a quantitative conversion of DTE derivatives in a crystalline state despite numorous molecular crystals of DTE derivatives having been synthesized. Kobatake et al.[39] prepared relatively thin crystals and achieved high photochemical conversion of the DTE derivative in the crystal state (>95%). However, as the photoreaction overcame the steric hindrance in a densely packed molecular crystal, the crystal showed a macroscopic deformation including contraction, bending and even fracture, which made the crystal structural transformation irreversible. In contrast, upon the transformation from PCP **1** to **2** in this study, such macroscopic crystal changes were not found, indicating that sufficient space and flexbility for the photochemical reactions of the DTE moieties are important factors for quantitative and reversible photochemical reaction in solids.

To elucidate the contribution of the flexible nature of the interpenetrated framework to the effective photochemical reaction in the solid, we conducted single-crystal X-ray diffraction experiments for photoirradiated crystals. A single-crystal of **1** was irradiated by UV light (365 nm UV-LED) for 30 min at room temperature, affording a single-crystal of photoirradiated PCP **2** ([Zn$_2$(bdc)$_2$(**L$_C$**)]·DMF)$_n$) through a SCSC transformation (Fig. 2g–i and Supplementary Fig. 2g–i, and Supplementary Table 1). First, a closed isomer **L$_C$** was crystallographically confirmed in **2**, with a C-C bond distance of 1.53 Å, suggesting the C-C bond formation by UV irradiation (Fig. 2g). While the global connectivity of the coordination bonds is maintained even after light irradiation (Fig. 2h and Supplementary Fig. 2g), one of the two pyridyl groups of **L$_C$** is inclined more to the 2D sheets composed of Zn$^{2+}$ and bdc$^{2-}$, thus providing a lateral sliding movement with a decrease in the interlayer distance (21.6 Å in **1** to 20.8 Å in **2**) (Fig. 2b, h). The void volume was significantly reduced from 30 to 21% with the decrease in the amount of guest molecules accommodated in the channels, which was also confirmed by TGA (Supplementary Fig. 6). Interestingly, the channels parallel to the *a*-axis that are decorated with the photochromic ligands in **1** were partially disconnected upon photoirradiation, and only a zig-zag 1D channel running parallel to the *a*-axis was observed in **2** (Fig. 2i and Supplementary Fig. 3c), as in the case of **1′** (Fig. 2f and Supplementary Fig. 3b). These results indicate that the molecular structural transformations of the DTE moieties from photochemical reaction induce the change of local coordination structure, resulting in the global structural changes of the entangled frameworks and the channels. These results motivated us to conduct sorption studies for the photoreactive porous crystals.

**Photomodulation of CO$_2$ sorption**. Photochemical control of gas sorption was then investigated for PCPs **1′** and **2′**

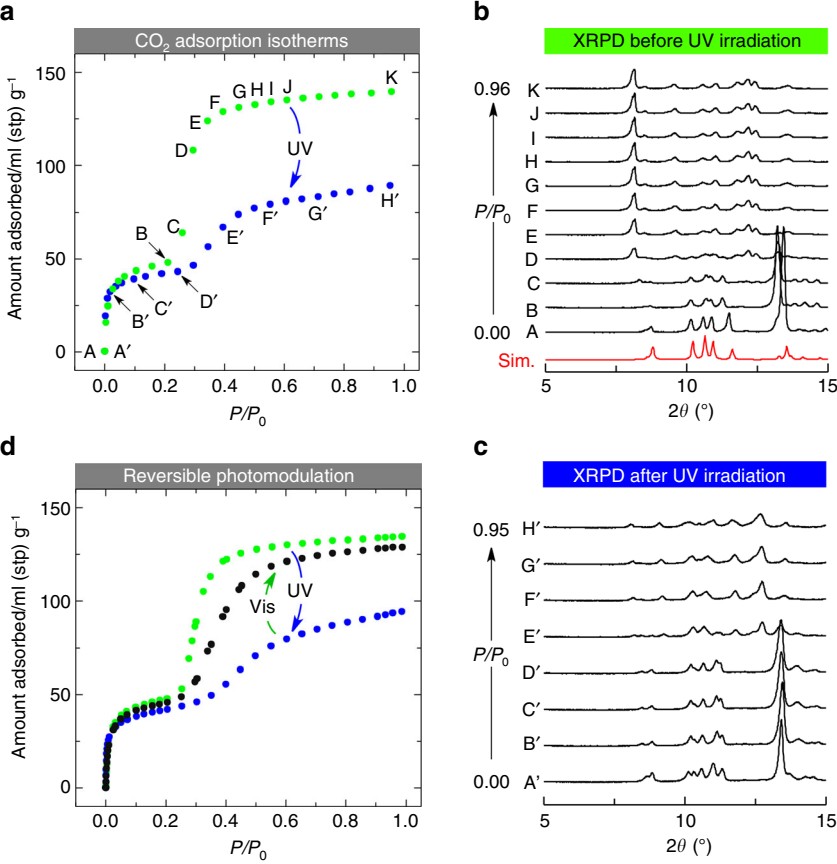

**Fig. 3** Coincident $CO_2$ adsorption and XRPD measurements. **a** $CO_2$ adsorption isotherms for non-irradiated **1'** (*green*) and UV-irradiated **2'** (*blue*) at 195 K. Stp means standard temperature and pressure. **b** XRPD patterns measured at each point (A to K) in the $CO_2$ adsorption isotherm of **1'**. A *red pattern* is the simulated one obtained from the single-crystal X-ray structure of **1'**. **c** XRPD patterns measured at each point (A' to H') in the $CO_2$ adsorption isotherm of **2'**. **d** Adsorption isotherms for the PCPs before irradiation (*green*), after UV irradiation (*blue*), and after successive irradiation with UV and visible light (*black*)

(Supplementary Figs. 5 and 7) using coincident X-ray powder diffraction (XRPD)/sorption measurement equipment[40, 41] with $CO_2$ as a molecular probe that has been utilized for the study of the flexibility in porous crystals[42, 43] (Fig. 3 and Supplementary Fig. 10). The sorption isotherms for both PCPs at 195 K showed clear steps (Fig. 3a). At the first steps in the adsorption branches, both PCPs **1'** and **2'** showed similar uptakes of $CO_2$ (48 and 43 ml (stp) g$^{-1}$ for **1'** and **2'**, respectively), without significant profile changes in XRPD patterns (**A** to **B** in Fig. 3b and **A'** to **D'** in Fig. 3c), which indicated that both PCPs have intrinsic microporous structures. In contrast, both PCPs showed marked changes in XRPD patterns after the flex points (**B** and **D'** in Fig. 3a for **1'** and **2'**, respectively), and 92 and 46 ml (stp) g$^{-1}$ of additional $CO_2$ were adsorbed up to $P/P_0 = 0.95$ in **1'** and **2'**, respectively, clearly indicative of the flexible nature of the twofold interpenetrated frameworks with a gate-opening process. The decrease in the total amount of $CO_2$ adsorbed in **2'** is reasonable as evidenced by the single-crystal structural analyses showing the void volume change (**1**: 30%, **2**: 21%). In the desorption processes, the PCPs released $CO_2$ in stepwise manner again and finally showed quite similar XRPD patterns to those before sorption (Supplementary Fig. 10). These results demonstrated that the structural transformations of the interpenetrated frameworks were fully reversible upon $CO_2$ sorption.

We also examined the reversibility of photomodulated $CO_2$ adsorption upon UV and visible light irradiation (Fig. 3d and Supplementary Fig. 11). The amount of adsorbed $CO_2$ at $P/P_0 = 0.98$ for non-irradiated PCP **1'** reached 136 ml (stp) g$^{-1}$.

After UV irradiation, the adsorption at $P/P_0 = 0.98$ for **2'** decreased to 108 ml (stp) g$^{-1}$. The adsorption recovered to 129 ml (stp) g$^{-1}$ after visible light irradiation. When we applied further UV irradiation again, the $CO_2$ adsorption dropped to 96 ml (stp) g$^{-1}$. As illustrated above, the reversible modulation of $CO_2$ sorption by photoirradiation is realized by the photoreactive flexible PCP. Our study not only clearly showed the correlation between photoisomerization efficiency and porous properties, which was not reported for pioneering works[22, 23] with a photomodulable $CO_2$ sorption, but also enables a material designed from the viewpoint of structural flexibility in a crystalline state. Our preliminary results in sorption experiments of PCP **1'** with other gases such as $N_2$ (Supplementary Fig. 12) indicated that the porous crystal is likely to show flexibility in sorption of various gases at different temperatures, and further investigation is required in the future.

## Discussion

Reversible photomodulation has been realized for various material properties such as photomagnetization[44] and photomechanical motions[45]. We successfully demonstrated highly reversible photomodulation of $CO_2$ sorption in a twofold interpenetrated photoreactive PCP containing a DTE-based ligand. The key to success is to introduce flexibility derived from the local and global structural changes in an entangled framework. Flexible crystalline porous frameworks can be synthesized using a wide variety of photoreactive species, including azobenzene[46], spiropyrane[47], and sterically hindered alkenes[48], which cannot be

converted between isomers in high yields in conventional solid materials because of their drastic conformation changes. Our strategy will grant access to a new dimension of porous compounds as platforms for various photochemical conversions and the photomodulation of porous properties.

## Methods

**Measurements**. $^1H$ NMR spectra were measured on a JEOL JNM-ECS 400 spectrometer using DMSO-$d_6$ or CDCl$_3$ as a solvent. Chemical shifts were referenced to the solvent values of 2.49 and 7.26 ppm for DMSO-$d_6$ and CDCl$_3$. Electrospray ionization time-of-flight (ESI-TOF) mass spectral measurements were performed using a Bruker micrOTOF II. Thermogravimetric analyses were recorded on a Rigaku Thermo plus Evo II TG-8120 apparatus in the temperature range between 30 and 500 °C under a nitrogen atmosphere at a heating rate of 5 °C min$^{-1}$. Single-crystal X-ray diffraction data collection was carried out on a Rigaku mercury diffractometer with a MoK$_\alpha$ radiation ($\lambda = 0.71069$ Å) and a RIGAKU Saturn70 CCD detector. XRPD data were collected with a Rigaku Ultima IV with CuK$_\alpha$ radiation ($\lambda = 1.5405$ Å). The sorption isotherm measurements for CO$_2$ were performed using an automatic volumetric adsorption apparatus (BELSORP-18PLUS; Bel Japan, Inc.) connected to a cryostat system. The coincident XRPD/sorption measurements were carried out on a Rigaku UltimaIV with CuK$_\alpha$ radiation connected to a BELSORP-18PLUS volumetric adsorption equipment (Bel Japan, Inc.).

**Photoirradiation**. A portable LED UV (ZUV-C20H; OMRON Corporation) was used to directly irradiate the single crystal of **1** fixed on the tip of the loop before measuring single-crystal X-ray diffraction of **2** to avoid visible light in the process of operation. A photochemical reactor (LZC-4; Luzchem) with a UV-lamp (365 or 306 nm) was used to irradiate a large amount of crystals for $^1H$ NMR and gas sorption measurements; the heat generated was controlled by a blowing fan. A green light (HDA-TB3; Hayashi Watch-Works Co., Ltd.) (530 nm) as a visible light source was applied to conduct the ring-opening process from **L$_C$** to **L$_O$** in the crystals. No light filter was employed in any photoirradiation process. For the reversible photomodulation of CO$_2$ adsorption shown in Fig. 3, powdery crystalline sample of **1′** (~30 mg) after the first CO$_2$ sorption experiment was irradiated with UV light for 24 h. During the irradiation period, to improve the irradiation efficiency on the bulk sample, the solid sample was agitated every two hours by a spatula manually. The conversion from **1′** to **2′** was confirmed as ~80% (**L$_C$/L$_O$** = 80/20) after 24 h irradiation. Then, the second sorption experiment was conducted. After the second sorption experiment, a similar procedure was applied to perform a backward reaction by irradition with visible light for 48 h, affording **L$_C$/L$_O$** = 10/90 in the solid substance. A similar procedure was applied again to achieve the solid sample for the fourth CO$_2$ sorption experiment. The ratio **L$_C$/L$_O$** in the sample was evaluated to be 10/90. All the sorption isotherms and the ratios of **L$_C$/L$_O$** are shown in Supplementary Fig. 11.

**Materials**. 1,2-Dichlorocyclopentane, $n$-butyl lithium solution, and tris(dibenzylideneacetone) dipalladium(0)-chloroform adduct were purchased from Sigma-Aldrich Co., Ltd. Cesium fluoride and tetrakis(triphenylphosphine)palladium were purchased from Tokyo Chemical Industry. 3-Bromo-2-methylthiophene, acetonitrile, and 4-pyridylboronic acid were purchased from Wako Pure Chemical Industries, Ltd. Triisopropyl borate, $N$-bromosuccinimide, acetic acid, ethyl acetate, hexane, methanol (MeOH), Na$_2$CO$_3$, NaOH and MgSO$_4$ were purchased from Nacalai Tesque. Tetrahydrofuran (THF) and DMF were purified by utilizing a Glass Contour solvent dispending system. Other reagents were reagent grade and used without further purification. 2-Methyl-3-thienylboronic acid and non-substituted DTE (compound **1**) was synthesized according to the procedures of Shinokubo et al.[49] 1,2-Bis(2′-methyl-5′-bromothien-3′-yl)perfluorocyclopentene (compound **2**) was prepared from compound **1** by reacting with $N$-bromosuccinimide according to the method reported by Park et al.[50]

**Synthesis of 1,2-bis(2′-methyl-5′-(4-pyridyl)thien-3′-yl)perfluorocyclopentene (L$_O$)**. Compound **2** (2.0 g, 3.8 mmol), 4-pyridylboronic acid (1.9 g, 15.2 mmol), and tetrakis(triphenylphosphine) palladium(0) (1.8 g, 0.15 mmol) were dissolved in a mixed solvent of THF (48 ml) and water (24 ml). The mixture was stirred for 24 h at 85 °C. The resulting solution was poured into ice-cold water, extracted with ethyl acetate, and dried over MgSO$_4$. The solvent was evaporated under reduced pressure and the residue was subjected to silica gel column chromatography using a mixture of MeOH/ethyl acetate as an eluent, affording a white solid substance. Yield: 1.4 g, 69%. $^1H$ NMR (400 MHz, CDCl$_3$, 30 °C) $\delta$ 8.612 (dd, $J = 4.6$ and 1.6 Hz, 4H, pyridyl), 7.486 (s, 2H, thienyl), 7.432 (dd, $J = 4.6$ and 1.6 Hz, 4H, pyridyl), 2.009 (s, 6H, methyl). ESI-MS: $m/z = 523.08$, calcd. for $(C_{25}H_{16}F_6N_2S_2)^+ = 523.07$ ($[M + H]^+$).

**Preparation of PCP 1**. 1,4-Benzenedicarboxylic acid (H$_2$bdc) (5 mM, 18 mg), Zn (NO$_3$)$_2$·6H$_2$O (5 mM, 29.7 mg), and **L$_O$** (200 mg, 20 mM) were dissolved in DMF. The solution was sealed in a glass vial and heated at 100 °C for 1 week. The crystals

obtained were washed with DMF and dried in vacuo at room temperature to give PCP **1**. Yield: 45 mg, 20%.

**Preparation of PCP 1′**. PCP **1** was dried under vacuum at 120 °C for 1 day to afford PCP **1′**.

**Preparation of PCP 2**. PCP **2** was obtained by irradiating PCP **1** by UV light for 1 day.

**Preparation of PCP 2′**. Similar to PCP **2**, PCP **2′** was obtained by irradiating PCP **1′** by UV light for 1 day.

**Data availability**. The X-ray crystallographic coordinates for structures reported in this study have been deposited at the Cambridge Crystallographic Data Centre (CCDC), under deposition numbers 1539241–1539243. These data can be obtained free of charge from The Cambridge Crystallographic Data Centre via www.ccdc.cam.ac.uk/data_request/cif.

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

## Acknowledgements

This work was supported by JSPS KAKENHI Grant-in-Aid for Specially Promoted Research (Grant No. 25000007) and ACCEL (Grant No. JPMJAC1302) project of the Japan Science and Technology Agency (JST). This study was also supported by JSPS KAKENHI Grant No. JP17H05357 (Coordination Asymmetry). iCeMS is supported by the World Premier International Research Initiative (WPI) of the Ministry of Education, Culture, Sports, Science, and Technology, Japan (MEXT). R.M. thanks JST-PRESTO program (Grant No. JPMJPR141C) for the financial support.

## Author contributions

Y.Z., H.S., and S.K. conceived and designed the project. Y.Z. and H.S. prepared and analyzed all compounds and carried out the sorption, spectroscopic measurements and photochemical experiments. R.M. supported the in situ XRPD/sorption measurements. P.W. and H.J.J. assisted the crystallographic study. Y.Z., H.S., and S.K. co-wrote the paper. All the authors discussed the results and commented on the manuscript.

## Additional information

**Competing interests:** The authors declare no competing financial interests.

