## [Peer Review File · Nature Communications]

Reviewers' comments:

Reviewer #1 (Remarks to the Author):

The work by Kitagawa reports on the synthesis and characterization of a novel photo-switchable MOF based on a flexible PCP. Such materials with photoswitchable porosity could be highly attractive to modulate separation and storage with light, or to develop photoswitchable catalysts and electronic devices, but are also fascinating in nature.

S. Kitagawa is among the leading scientists in the MOF/PCP field and his work is of high scientific quality. The materials are thoroughly characterized by single crystal XRD and in situ powder XRD following the structural changes during CO₂ adsorption.

The authors clearly demonstrate photoswitchable porosity by single crystal X-ray structure analysis using a flexible MOF architecture and a dithienylethene (DTE) derivative linker, a moiety well known for molecular switchability, which has been reported before and also integrated into MOF architectures, different from the ones reported here, by Shustova, however, without demonstrating photoswitchable porosity.

In the work by Kitagawa, for the first time optical porosity switching in a MOF is shown. The changes in the CO₂ isotherms after irradiation are significant and repeatable. After irradiation, the material changes its flexibility characteristics resulting in an overall lower CO₂ uptake.

Interestingly the first step in the CO₂ isotherms is not affected which is consistent with the crystallographic results. Following the intermediates via in situ PXRD gives information about the different structural trajectories and is quite interesting. It may be also recognized that the crystallinity of the open form after irradiation shows broader peaks indicating a more defective structure. Nevertheless, repeated structure transformation is shown. While it is reasonable to use CO₂ for demonstrating in situ the structural transformations it is not so clear if the adsorption of other gases will be affected in a similar or totally different way. The choice of CO₂, which is a probe for very small pores, could be explained more detailed.

Overall this is an excellent and valuable contribution to Nature Communications.

Minor comments:

1) The work by N. Shustova should be cited

2) The labelling of Fig. 1 should be expanded:

- The scale is missing in the fotos
- The numbers are missing in the isotherms (x,y)
- The temperature for the CO₂ adsorption experiment is missing

3) P5: "isomerimzation" (typo)

Reviewer #2 (Remarks to the Author):

This paper describes a metal-organic framework (MOF) that is built up using a photochromic diarylethene-pyridyl ligand. Triggered by light, this ligand is able to switch between open and closed forms in the MOF. This switching is reversible (broadly speaking) and it reduces the CO₂ uptake capacity of the MOF by around 50%.

A number of major claims are made in the manuscript. I will address several of these in points 1-4 below:

1. In the abstract is stated that "quantitative conversion in photochemical reactions in solid is

hampered by the lack of intrinsic structural flexibility for releasing stress and strain upon photochemical events." This statement is not followed up on in the main text. A literature review serving justify this statement would put the rationale behind the paper on a much firmer footing. 2. A "new stratgey" towards photochromic frameworks is a justifiable claim. This strategy involves the use of a type of framework that is known to be flexible. This assists the photoisomerization process, which involves a significant contraction of the ligand. This is the most significant feature of the manuscript. I do note, however, that the ligand design is not novel, having been reported in ref 19.

3. The authors that that the photoswitching in the MOF is "quantitative" and "reversible". It is nearly quantitative (96-97%) but the reversibility is questionable since no appropriate experimental data are reported (e.g. NMR spectra of dissolved samples after the reverse photoisomerization back to the open form). The CO₂ adsorption isotherms are indirect evidence, but the clear discrepancy between the isotherm of the original framework and the that after visible irradiation in Fig. 3d suggest a significant degree of irreversibility.

4. The manuscript speaks of the photoisomerization being a "single-crystal-to-single-crystal (SCSC)" transformation. However, this is not justified or even explicitly discussed in the main text. There are significant changes in the crystallographic parameters (Table S1) which should be fully teased apart in the manuscript.

5. Other shortcomings of the manuscript that came to my attention include:

(i) The manuscript is generally light on detail and seems have been rushed into submission without a comprehensive analysis of the results and the development of an informed set of discussion points. An example of this is that the C-C bond length of the isomerized diarylethene is only touched upon in a figure and not discussed in the main text.

(ii) The English is idiosyncratic and could be significantly improved.

(iii) The experimental section should detail the set-up used for the photolysis reactions.

(iv) That the ligand has been previously reported should be noted by appropriate citations in the main text and the ESI.

On the basis of the foregoing comments unfortunately I cannot recommend this paper for publication in Nature Communications. It does have some merit in that the flexible framework appears to raise the efficiency of the photoisomerisation. However, in many other aspects this work falls short of the required standard for publication in a high-impact forum.

Reviewer #3 (Remarks to the Author):

The authors describe the design and synthesis of an interpenetrated metal-organic framework that incorporates a photoresponsive diarylethene unit into the linkers of the MOF. They have achieved an apparently quantitative and reversible photoconversion of the linkers within the framework, with the photoswitched state demonstrating reduced CO₂ uptake in comparison with the original state.

The major novelty in this work is in the use of an interpenetrated framework, which imparts sufficient flexibility on the overall structure that complete photoconversion between states is possible. Given the difficulty in the photocrystallographic community in achieving high levels of photoconversion (e.g. of linkage isomers in molecular species), this strategy of exploiting framework interpenetration complements the limited existing literature on photoswitching stabilised by coordination polymers and MOFs and adds another tool to the community's arsenal. It is particularly pleasing to see that the photoswitched state is crystalline and ordered, allowing the authors to obtain the crystal structure of this species, something that is currently uncommon in other examples (e.g. doi: 10.1002/anie.201206359).

There are a few minor points to address (below), but on the whole this is an excellent piece of work of significant interest in the field of responsive porous materials, and is certainly worthy of rapid publication in Nature Communications.

1. Topology:

The use of interpenetration is very successful in achieving high photoconversion of the DTE-based linkers. Was this intentional, or a serendipitous discovery? How important is the topology of the nets in achieving this and was that a consideration in the linker design? Can the authors comment on the topological constraints of their approach in future MOF design? Clearly the structure changes on guest removal are important indicators of the MOF flexibility and are very well described as such in the manuscript.

2. References and broader context:

The references are an interesting selection, spanning work from a wide range of groups active in the field. It is a little surprising that there are some omissions in what is still a fairly small niche area of the metal-organic framework literature. Photoresponsive MOFs have been ably reviewed in several recent papers (e.g. (doi): 10.1039/C5CE02543E, 10.1039/C5TA09424K, 10.1021/acs.chemmater.5b00046 for three examples), which can be used to cover the omissions. Given the use of diarylethenes I would also expect more of a spotlight on and more direct comparison with i) the work of Benedict's group in Buffalo, US, who have reported a number of frameworks that are relevant to the present manuscript (<http://www.benedictresearchlabs.com/>), and to which there appears to be only a single reference, and ii) the CO₂ uptake photomodulation in the important work of Guo et al. which is referenced in the current manuscript as ref 19. It is noteworthy that the authors' material appears to have higher structural photoconversion than that in ref 19, but lower desorption capacity than both Guo's framework (~75%) and the seminal example of Hill et al. (doi 10.1002/anie.201206359, not referenced in the current manuscript) using azobenzene linkers (64%). This should be discussed, as it has implications in the future design of materials for such gas-uptake switching applications.

3. Experimental details:

- a. The y-axis label on Figure S7 must be incorrect. It states "Intensity" when an absorption spectrum is shown. Is this instead F(R) from measuring reflectance and converting using the Kubelka-Munk function?
- b. The preparation details for PCP 2 and 2' are inadequate. In what setup were the samples irradiated, with what source (specifically, what wavelengths are important and what power and type of light source was used?), how was temperature controlled (if a heat-producing UV source was used), was the light filtered in any way...? The details in the experimental section do not agree with the main text on Page 9, which describes illumination with a 365 nm LED (just one? What power?) for only 30 minutes; the experimental section simply states that UV irradiation was used for 1 day (24 hours precisely? How was conversion % measured?).
- c. What was the visible light source used to perform the photomodulation of CO₂ uptake? How was the photomodulation gas sorption experiment performed – was it in-situ in the volumetric adsorption apparatus? This needs to be described, and if the measurement was performed in-situ in the BELSORP-18PLUS, details of the equipment and geometry used to accomplish this would be greatly appreciated. This will aid other groups in performing directly comparable experiments and in reproducing the authors' approach in the future.

4. Photochemistry:

Given this is a communication, it is acceptable not to go into more details about the photoprocess itself. However, in a full paper, I would expect better information about the absorption and photoconversion behaviour of 1 and 1'. Example questions still to be answered include: How efficient is the photoconversion? What is the mechanism of propagation through single crystals? What is the dependence on wavelength? How rapid is the back-conversion process? If materials like these are to ever be employed for their gas uptake photomodulation behaviour then these sorts of questions become very important, and they are not addressed in the current work.

5. A comment:

Given the nature of this communication and the desirability of rapid publication of these important findings, it is understandable that more information is not available about the structure of the materials after the second step in the adsorption isotherms of CO₂ has been passed. I eagerly look

forward to seeing the future results of in-situ gas-loading crystallographic experiments that are crucial to determining exactly what is happening structurally in these crystals above the CO₂ gate pressure. The changes in the PXRD pattern are, as noted, significant, and hint at the key important differences in the behaviour of the material before vs after photoswitching, and during the hysteresis in the isotherms.

Point-by-point response to the referees' comments

The reviewers' comments are written with *Italic font* and replies are written in **blue**.

Reply to the Reviewer #1's comments

The work by Kitagawa reports on the synthesis and characterization of a novel photo-switchable MOF based on a flexible PCP. Such materials with photoswitchable porosity could be highly attractive to modulate separation and storage with light, or to develop photoswitchable catalysts and electronic devices, but are also fascinating in nature. S. Kitagawa is among the leading scientists in the MOF/PCP field and his work is of high scientific quality. The materials are thoroughly characterized by single crystal XRD and in situ powder XRD following the structural changes during CO₂ adsorption. The authors clearly demonstrate photoswitchable porosity by single crystal X-ray structure analysis using a flexible MOF architecture and a dithienylethene (DTE) derivative linker, a moiety well known for molecular switchability, which has been reported before and also integrated into MOF architectures, different from the ones reported here, by Shustova, however, without demonstrating photoswitchable porosity.

In the work by Kitagawa, for the first time optical porosity switching in a MOF is shown. The changes in the CO₂ isotherms after irradiation are significant and repeatable. After irradiation, the material changes its flexibility characteristics resulting in an overall lower CO₂ uptake. Interestingly the first step in the CO₂ isotherms is not affected which is consistent with the crystallographic results. Following the intermediates via in situ PXRD gives information about the different structural trajectories and is quite interesting. It may be also recognized that the crystallinity of the open form after irradiation shows broader peaks indicating a more defective structure. Nevertheless, repeated structure transformation is shown.

While it is reasonable to use CO₂ for demonstrating in situ the structural transformations it is not so clear if the adsorption of other gases will be affected in a similar or totally different way. The choice of CO₂, which is a probe for very small pores, could be explained more detailed.

CO₂ is a suitable and popular probe for the investigation of sorption properties of flexible frameworks as reported previously (e.g. Seo *et al.* *J. Am. Chem. Soc.* **133**, 9005–9013 (2011); Zhou *et al.* *Nature Commun.* **4**, 2534 (2013)). It is known that interpenetrated frameworks often show kinetically non-porous nature for N₂ due to its slow diffusion process in micropores at low temperature (77 K). In order to eliminate such concern this time we decided to use CO₂ as a probe originally. However, N₂ is also a commonly used to study the flexible nature, so we measured N₂ sorption isotherms on PCP **1'** (Supplementary Fig. 12). For N₂, a stepwise adsorption isotherm was likewise observed for PCP **1'** at 77 K, indicating that the porous crystal is likely to show flexibility in sorption of various gases at different temperatures, and further investigation is required in the future. Accordingly, we revised our manuscript as follows.

Page 10, line 19: “*Our preliminary results in sorption experiments of PCP 1' with other gases such as N₂ (Supplementary Fig. 12) indicated that the porous crystal is likely to show flexibility in sorption of various gases at different temperatures, and further investigation is required in the future.*”

Supplementary Figure 12. Gas sorption isotherms of PCP **1'** for CO₂ (195 K) and N₂ (77 K).

In addition, for better understanding of broad readers, two references were added as ref. 42 and 43, where distinct sorption behaviours were observed in flexible porous crystals for CO₂ and N₂, and we revised the manuscript as follows.

Page 9, line 10: “Photochemical control of gas sorption was then investigated for PCPs **1**’ and **2**’ using coincident XRPD/sorption measurement equipment^{40,41} with CO₂ as a molecular probe that has been utilized for the study of the flexibility in porous crystals^{42,43}.”

Overall this is an excellent and valuable contribution to Nature Communications.

We really appreciate the reviewer’s encouraging remarks.

Minor comments:

1) The work by N. Shustova should be cited.

The work by Shustova *et al.* reported in *JACS* was cited as ref. 23.

2) The labelling of Fig. 1 should be expanded:

- The scale is missing in the fotos

The scale bars were added in Figure 1 and the caption was revised.

- The numbers are missing in the isotherms (x,y)

The isotherms with the values are shown in the revised figure.

- The temperature for the CO₂ adsorption experiment is missing

The temperature (195 K) for the CO₂ adsorption was added in the figure caption.

3) P5: “isomerimzation” (typo)

The typo was corrected as “isomerization” in page 4, line 20.

Reply to the Reviewer #2's comments

This paper describes a metal-organic framework (MOF) that is built up using a photochromic diarylethene-pyridyl ligand. Triggered by light, this ligand is able to switch between open and closed forms in the MOF. This switching is reversible (broadly speaking) and it reduces the CO₂ uptake capacity of the MOF by around 50%.

A number of major claims are made in the manuscript. I will address several of these in points 1-4 below:

- 1. In the abstract is stated that "quantitative conversion in photochemical reactions in solid is hampered by the lack of intrinsic structural flexibility for releasing stress and strain upon photochemical events." This statement is not followed up on in the main text. A literature review serving justify this statement would put the rationale behind the paper on a much firmer footing.*

*We really appreciate the reviewer's comments and added some discussions (page 7, line 19) on a previous report by Kobatake *et al.* about macroscopic deformation of a crystal, which might be related to flexibility in the solid.*

- 2. A "new stratgey" towards photochromic frameworks is a justifiable claim. This strategy involves the use of a type of framework that is known to be flexible. This assists the photoisomerization process, which involves a significant contraction of the ligand. This is the most significant feature of the manuscript. I do note, however, that the ligand design is not novel, having been reported in ref 19.*

*As pointed out by the reviewer, the ligand itself was reported as the component of a MOF by Luo *et al.* (*Angew. Chem. Int. Ed.* **53**, 9298–9301 (2014).) Before the paper, the compound was firstly reported by Lehn's group more than 20 years ago (*J. Chem. Soc., Chem. Commun.* 1439–1442 (1993)) and have been utilized as a molecular components of photochromic materials including coordination polymers (*Inorg. Chem.* **43**, 482–489 (2004); *Inorg. Chem.* **43**, 3774–3776 (2004)). However, to our surprise, there was no report on a quantitative analysis of the photochemical conversion of the DTE ligand in the solid. In our report, we dared to choose such a reported ligand without modification on the ligand itself to show that our strategy is applicable to a typical DTE ligand.*

3. *The authors that that the photoswitching in the MOF is "quantitative" and "reversible". It is nearly quantitative (96-97%) but the reversibility is questionable since no appropriate experimental data are reported (e.g. NMR spectra of dissolved samples after the reverse photoisomerization back to the open form). The CO₂ adsorption isotherms are indirect evidence, but the clear discrepancy between the isotherm of the original framework and the that after visible irradiation in Fig. 3d suggest a significant degree of irreversibility.*

First, we confirmed the reversibility in its photoswitching phenomena by our naked eyes as shown in Fig. 1. The deep blue UV-irradiated crystal was returned back to a colorless crystal after visible light irradiation, indicating its quantitative reaction from close to open isomer in the crystals. We also confirmed its reversibility using ¹H NMR spectroscopy as suggested by the reviewer and added the spectra as Supplementary Fig. 9 in the revised Supplementary Information. The experimental procedure is as follows. After 1-h UV irradiation of PCP **1** (~5 mg) sandwiched by two glass plates, a half (2~3 mg) of the irradiated sample was digested in DMSO-*d*₆/aq. HCl for ¹H NMR analysis of the mixtures containing the open and close isomers. The NMR analysis revealed that the conversion from open- to close isomer in the PCP was 97% ($L_C/L_O = 97/3$). After 12-h visible light irradiation of the UV-irradiated solid substance (2~3 mg) sandwiched by two glass plates, the conversion for the backward reaction from L_C to L_O in the PCP was estimated as 95% ($L_C/L_O = 5/95$). These results clearly showed that the photochemical reactions in the PCP are almost quantitative and reversible intrinsically. Although the quantitative and reversible reactions in both directions were easily achieved in a relatively small scale (~5 mg of the PCP), we found that the bulk amount of the crystals (~50 mg) was difficult to be converted in a similar efficiency because of a difficulty in homogeneous irradiation of the solid substance in a sample cell for the sorption experiment, which might be the reason for the discrepancy between the isotherm of the original framework and that after visible irradiation ($L_C/L_O = 10/90$ after 48-h visible light irradiation) in Fig. 3. We have added "Photoirradiation" section in page 12, and the information about the ratios of L_C/L_O in the solid substances for the sorption experiments in Supplementary Fig. 11.

Supplementary Figure 9. Reversible photochemical conversions between L_O and L_C in the PCP crystals. 1H NMR spectra of guest-free 1' (a) before and (b) after irradiation with UV (PCP 2'), and (c) after irradiation with UV and visible light (recovered PCP 1'). The solid substances were digested in $DMSO-d_6/aq. HCl$ for 1H NMR spectroscopy. The ratios of L_C/L_O in PCP 2' and recovered PCP 1' are estimated to be 97/3 and 5/95, respectively. The signals marked with an asterisk are assignable to those of the solvents and water. A schematic illustration for the preparation of the samples for 1H NMR spectroscopy is shown on the top.

4. The manuscript speaks of the photoisomerization being a "single-crystal-to-single-crystal (SCSC)" transformation. However, this is not justified or even explicitly discussed in the main text. There are significant changes in the crystallographic parameters (Table S1) which should be fully teased apart in the manuscript.

According to the reviewer's comments, we revised the section "Photochemical transformation of a soft porous crystal in an SCSC manner" in the main text (page 8, line 10) and the experimental section in "Methods" was also revised.

5. Other shortcomings of the manuscript that came to my attention include:

(i) The manuscript is generally light on detail and seems have been rushed into submission without a comprehensive analysis of the results and the development of an informed set of discussion points. An example of this is that the C-C bond length of the isomerized diarylethene is only touched upon in a figure and not discussed in the main text.

We carefully revised the manuscript to discuss the details of the structural transformation based on the figures. In fact, we added the information about the C-C bond length as follows.

Page 8, line 12: "First, a closed isomer L_C was crystallographically confirmed in **2** with a C-C bond distance of 1.53 Å, indicating the C-C bond formation by UV irradiation (Fig. 2g)."

(ii) The English is idiosyncratic and could be significantly improved.

According to the reviewer's comment, we have carefully revised our manuscript.

(iii) The experimental section should detail the set-up used for the photolysis reactions.

We have added a section of "Photoirradiation" in "Methods" in page 12 to detail the setup for the photochemical reactions.

(iv) That the ligand has been previously reported should be noted by appropriate citations in the main text and the ESI.

As pointed out by the reviewer, the ligand as the component of a MOF was reported by Luo *et al.* (*Angew. Chem. Int. Ed.* **53**, 9298–9301 (2014)). Although the paper was already cited as ref. 19 in the initial manuscript, we have cited the paper appropriately

in the revised manuscript. In addition, some related papers including one (*J. Chem. Soc., Chem. Commun.* 1439–1442 (1993)) which firstly reported the compound has been cited. The cited papers related to the ligand are shown in the following list.

The first report for the synthesis of the compound.

36. Gilat, S. L., Kawai, S. H. & Lehn, J.-M. Light-triggered electrical and optical switching devices. *J. Chem. Soc., Chem. Commun.* 1439–1442 (1993).

The first report for the coordination polymer containing the ligand.

37. Matsuda, K., Takayama, K. & Irie, M. Photochromism of metal complexes composed of diarylethene ligands and Zn(II), Mn(II), and Cu(II) hexafluoroacetylacetonates. *Inorg. Chem.* **43**, 482–489 (2004).

The first report for the MOF composed of the ligand.

22. Luo, F. *et al.* Photoswitching CO₂ capture and release in a photochromic diarylethene metal–organic framework. *Angew. Chem. Int. Ed.* **53**, 9298–9301 (2014).

On the basis of the foregoing comments unfortunately I cannot recommend this paper for publication in Nature Communications. It does have some merit in that the flexible framework appears to raise the efficiency of the photoisomerisation. However, in many other aspects this work falls short of the required standard for publication in a high-impact forum.

We appreciate the reviewer's inputs and have revised our manuscript according to the comments. We hope our revised manuscript is now suitable for publication in *Nature Communications*.

Reply to the Reviewer #3's comments

The authors describe the design and synthesis of an interpenetrated metal-organic framework that incorporates a photoresponsive diarylethene unit into the linkers of the MOF. They have achieved an apparently quantitative and reversible photoconversion of the linkers within the framework, with the photoswitched state demonstrating reduced CO₂ uptake in comparison with the original state. The major novelty in this work is in the use of an interpenetrated framework, which imparts sufficient flexibility on the overall structure that complete photoconversion between states is possible. Given the difficulty in the photocrystallographic community in achieving high levels of photoconversion (e.g. of linkage isomers in molecular species), this strategy of exploiting framework interpenetration complements the limited existing literature on photoswitching stabilised by coordination polymers and MOFs and adds another tool to the community's arsenal. It is particularly pleasing to see that the photoswitched state is crystalline and ordered, allowing the authors to obtain the crystal structure of this species, something that is currently uncommon in other examples (e.g. doi: 10.1002/anie.201206359).

There are a few minor points to address (below), but on the whole this is an excellent piece of work of significant interest in the field of responsive porous materials, and is certainly worthy of rapid publication in Nature Communications.

We really appreciate such encouraging remarks by the reviewer.

1. Topology:

The use of interpenetration is very successful in achieving high photoconversion of the DTE-based linkers. Was this intentional, or a serendipitous discovery? How important is the topology of the nets in achieving this and was that a consideration in the linker design? Can the authors comment on the topological constraints of their approach in future MOF design? Clearly the structure changes on guest removal are important indicators of the MOF flexibility and are very well described as such in the manuscript.

Before synthesizing our flexible porous crystals, we successfully obtained a different porous crystal composed of Zn²⁺, 4,4'-biphenyldicarboxylate, and a DTE-ligand (L₀), and the crystal structure was a **dia** net structure with five fold interpenetration (similar to the reported one by Luo *et al.* in *Angew. Chem. Int. Ed.* **53**, 9298–9301 (2014)).

Unfortunately, the photochemical reactivity of the crystal was found to be moderate [redacted] We assumed that its dense packing and the structural rigidity prevent the DTE moieties being rearranged upon photochemical reactions, resulting in a low conversion. Then, we decided to synthesize flexible frameworks allowing DTE moieties to show structural transformation necessary for high conversion in the photochemical reactions. Therefore, we can say our discovery was intentional. The net topology of **1** is primitive cubic (**pcu**), which is a well-known topology for interpenetrated frameworks (Jiang *et al. Coord. Chem. Rev.* **257**, 2232–2249 (2013)). In addition, through our previous researches about flexible porous crystals based on the same topology (*Angew. Chem. Int. Ed.* **49**, 7660–7664 (2010)), we have been ready to design such a flexible framework with a bispyridyl-type pillar.

[Redacted]

[Redacted]

We believe our strategy based on interpenetrated frameworks for highly effective photochemical reactions is applicable to various interpenetrated frameworks with different topologies. One of the important factor is the degree of flexibility. For example, even in interpenetrated frameworks, a control of interpenetration degree is quite important to achieve flexible nature in the solid [redacted]

[Redacted]

2. References and broader context:

The references are an interesting selection, spanning work from a wide range of groups active in the field. It is a little surprising that there are some omissions in what is still a fairly small niche area of the metal-organic framework literature. Photoresponsive MOFs have been ably reviewed in several recent papers (e.g. (doi): 10.1039/C5CE02543E, 10.1039/C5TA09424K, 10.1021/acs.chemmater.5b00046 for three examples), which can be used to cover the omissions.

Thank you for the suggestions. We have updated the references 10–12 with recent review papers as the reviewer suggested.

Given the use of diarylethenes I would also expect more of a spotlight on and more direct comparison with i) the work of Benedict's group in Buffalo, US, who have reported a number of frameworks that are relevant to the present manuscript (<http://www.benedictresearchlabs.com/>), and to which there appears to be only a single reference, and ii) the CO₂ uptake photomodulation in the important work of Guo et al.

which is referenced in the current manuscript as ref 19. It is noteworthy that the authors' material appears to have higher structural photoconversion than that in ref 19, but lower desorption capacity than both Guo's framework (~75%) and the seminal example of Hill *et al.* (doi 10.1002/anie.201206359, not referenced in the current manuscript) using azobenzene linkers (64%). This should be discussed, as it has implications in the future design of materials for such gas-uptake switching applications.

According to the reviewer's comments, we have added the references reported by Benedict *et al.* as ref. 20 and 21, and one paper reported by Hill *et al.* as ref. 16 in the revised manuscript. Although the papers reported by Guo's group and Hill's group are seminal as the reviewer mentioned, our work does not aim to merely improve the desorption capacity. Our results not only clearly showed the correlation between photoisomerization efficiency and porous properties, but also enable a new material design from the viewpoint of structural flexibility in a crystalline state.

We have added the following sentence in page 10, line 15. *"Our study not only clearly showed the correlation between photoisomerization efficiency and porous properties, which was not reported for pioneering works with a photomodulable CO₂ sorption, but also enables a new material designed from the viewpoint of structural flexibility in a crystalline state."*

3. Experimental details:

a. The y-axis label on Figure S7 must be incorrect. It states "Intensity" when an absorption spectrum is shown. Is this instead $F(R)$ from measuring reflectance and converting using the Kubelka-Munk function?

Thank you for the comment on the spectra. We replaced the spectra with diffuse reflectance spectra in Supplementary Fig. 7.

b. The preparation details for PCP 2 and 2' are inadequate. In what setup were the samples irradiated, with what source (specifically, what wavelengths are important and what power and type of light source was used?), how was temperature controlled (if a heat-producing UV source was used), was the light filtered in any way...? The details in the

experimental section do not agree with the main text on Page 9, which describes illumination with a 365 nm LED (just one? What power?) for only 30 minutes; the experimental section simply states that UV irradiation was used for 1 day (24 hours precisely? How was conversion % measured?).

We have added “Photoirradiation” section in page 12 to show the details for the sample preparation.

c. What was the visible light source used to perform the photomodulation of CO₂ uptake? How was the photomodulation gas sorption experiment performed – was it in-situ in the volumetric adsorption apparatus? This needs to be described, and if the measurement was performed in-situ in the BELSORP-18PLUS, details of the equipment and geometry used to accomplish this would be greatly appreciated. This will aid other groups in performing directly comparable experiments and in reproducing the authors’ approach in the future.

We appreciate the reviewer’s comments on the experimental setup for the photochemical modulation of CO₂ sorption. The photomodulation of the sorption was not performed in an *in-situ* manner. Powdery crystalline sample of **1**’ after the 1st CO₂ sorption experiment was irradiated with UV for 24 h. During the irradiation period, to improve the irradiation efficiency on the bulk sample, the solid sample was agitated every several hours by a spatula manually. The conversion from **1**’ to **2**’ was confirmed as ~80% ($L_C/L_O = 80/20$) after 24 h irradiation. Then, the 2nd sorption experiment was conducted. After the 2nd sorption experiment, a similar procedure was applied to perform a backward reaction by the irradiation with visible light for 48 h, affording $L_C/L_O = 10/90$ in the solid substance. A similar procedure was applied again to achieve the solid sample for the 4th CO₂ sorption experiment. The ratio L_C/L_O in the sample was evaluated to be 90/10. All the sorption isotherms and the ratios of L_C/L_O are shown in Supplementary Fig. 11. We have added the detailed procedure in the section of “Photoirradiation” in page 12.

4. Photochemistry:

*Given this is a communication, it is acceptable not to go into more details about the photoprocess itself. However, in a full paper, I would expect better information about the absorption and photoconversion behaviour of **1** and **1**’. Example questions still to be*

answered include: How efficient is the photoconversion? What is the mechanism of propagation through single crystals? What is the dependence on wavelength? How rapid is the back-conversion process? If materials like these are to ever be employed for their gas uptake photomodulation behaviour then these sorts of questions become very important, and they are not addressed in the current work.

We are very glad to know the reviewer's intense interests on the future directions of our research targets. All the aspects questioned by the reviewer are quite important to approach for further development of photoresponsive porous materials. We are also highly motivated to deepen the research field and have already performed some experiments. Our preliminary results about the photostationary state of the diarylethene derivatives showed the wavelength-dependent reactivity. UV light irradiation at different wavelength achieves the different ratio between open- and closed isomers at the photostationary state. We hope that the results will be reported in the future.

5. A comment:

Given the nature of this communication and the desirability of rapid publication of these important findings, it is understandable that more information is not available about the structure of the materials after the second step in the adsorption isotherms of CO₂ has been passed. I eagerly look forward to seeing the future results of in-situ gas-loading crystallographic experiments that are crucial to determining exactly what is happening structurally in these crystals above the CO₂ gate pressure. The changes in the PXRD pattern are, as noted, significant, and hint at the key important differences in the behaviour of the material before vs after photoswitching, and during the hysteresis in the isotherms.

We highly appreciate the reviewer's kind suggestions. Although we have not obtained the structural information about the CO₂-adsorbed phase, we are also quite interested in the relationship between the framework flexibility and the photochemical reactivity. The studies related to this aspect are ongoing in our group, and we hope that the results will be reported in the future.

REVIEWERS' COMMENTS:

Reviewer #1 (Remarks to the Author):

The authors have carefully addressed all comments of the referees and made the appropriate changes. Additional data were added and discussed in the main manuscript as well as the supporting information. Sorption data for nitrogen were added and discussed. Moreover, several references were added placing the work in the appropriate context.

In conclusion the revised manuscript is a valuable contribution to nature communications.

Reviewer #2 (Remarks to the Author):

The manuscript has been well revised in light of the reviewers' comments and is now suitable for publication.

Reviewer #3 (Remarks to the Author):

The authors have provided a thorough and well-considered response to all the referees' comments. They have certainly addressed all my concerns and therefore, in line with my original recommendation, I strongly support publication of this manuscript in Nature Communications.

Point-by-point response to the referees' comments

The reviewers' comments are written with *Italic font* and replies are written in **blue**.

Reviewer #1's comments

The authros have carefully addressed all comments of the referees and made the appropriate changes. Additional data were added and discussed in the main manuscript as well as the supportings. Sorption data for nitrogen were added and discussed. Moerover, several references were added placing the work in the appropriate context. In conclusion the revised mansucrypt is a vaulable contribution to nature communications.

We really appreciate the reviewer's comments.

Reviewer #2's comments

The manuscript has been well revised in light of the reviewers' comments and is now suitable for publication.

We appreciate the reviewer's inputs and understanding our efforts to improve our paper.

Reviewer #3's comments

The authors have provided a thorough and well-considered response to all the referees comments. They have certainly addressed all my concerns and therefore, in line with my original recommendation, I strongly support publication of this manuscript in Nature Communications.

We really appreciate such encouraging remarks by the reviewer.